

# Hepatic fibrosis and factors associated with liver stiffness in HIV mono-infected individuals

Mihály Sulyok[1,2], Tamás Ferenci[3], Mihály Makara[4,5], Gábor Horváth[5], János Szlávik[6], Zsófia Rupnik[6], Luca Kormos[6], Zsuzsanna Gerlei[7], Zita Sulyok[2] and István Vályi-Nagy[4,6]

[1] Doctoral School for Clinical Medicine, Semmelweis University, Budapest, Hungary
[2] Institute for Tropical Medicine, Eberhard Karls University, Tuebingen, Germany
[3] John von Neumann Faculty of Informatics, Physiological Controls Group, Óbuda University, Budapest, Hungary
[4] Center for Hepatology, St. István and St László Hospital, Budapest, Hungary
[5] Hepatology Center of Buda, Budapest, Hungary
[6] Center for HIV, St. István and St László Hospital, Budapest, Hungary
[7] Transplantation and Surgical Clinic, Semmelweis University, Budapest, Hungary

Corresponding author
Mihály Sulyok, mihaly.sulyok@uni-tuebingen.de

## ABSTRACT

**Background**. Liver disease has become an important cause of morbidity and mortality even in those HIV-infected individuals who are devoid of hepatitis virus co-infection. The aim of this study was to evaluate the degree of hepatic fibrosis and the role of associated factors using liver stiffness measurement in HIV mono-infected patients without significant alcohol intake.

**Methods**. We performed a cross-sectional study of 101 HIV mono-infected patients recruited prospectively from March 1, 2014 to October 30, 2014 at the Center for HIV, St István and St László Hospital, Budapest, Hungary. To determine hepatic fibrosis, liver stiffness was measured with transient elastography. Demographic, immunologic and other clinical parameters were collected to establish a multivariate model. Bayesian Model Averaging (BMA) was performed to identify predictors of liver stiffness.

**Results**. Liver stiffness ranged from 3.0–34.3 kPa, with a median value of 5.1 kPa (IQR 1.7). BMA provided a very high support for age (Posterior Effect Probability-PEP: 84.5%), moderate for BMI (PEP: 49.3%), CD4/8 ratio (PEP: 44.2%) and lipodystrophy (PEP: 44.0%). For all remaining variables, the model rather provides evidence against their effect. These results overall suggest that age and BMI have a positive association with LS, while CD4/8 ratio and lipodystrophy are negatively associated.

**Discussion**. Our findings shed light on the possible importance of ageing, overweight and HIV-induced immune dysregulation in the development of liver fibrosis in the HIV-infected population. Nonetheless, further controlled studies are warranted to clarify causal relations.

## INTRODUCTION

Liver disease has become one of the most important cause of morbidity and mortality in HIV-infected individuals (*Weber et al., 2006*). While hepatitis B or C co-infections remain the most important cause of liver damage, liver related mortality also affects those infected only with HIV (*Antiretroviral Therapy Cohort Collaboration, 2010*). Long term antiretroviral and non-antiretroviral medications, HIV induced long term inflammation, metabolic complications and direct cytopathic effects may also contribute to the pathogenesis of liver fibrosis (LF) (*Rockstroh et al., 2014*). An increasing number of papers have been published on fibrosis in HIV/hepatitis virus co-infected patients (*Audsley et al., 2016*; *Brunet et al., 2016*; *Costiniuk et al., 2016*; *Fernández-Montero et al., 2013*; *Gonzalez et al., 2015*; *Ioannou et al., 2015*; *Kliemann et al., 2016*; *Konerman et al., 2014*; *Kooij et al., 2016*; *Li Vecchi et al., 2013*; *Macías et al., 2013a*; *Macías et al., 2013b*; *Njei et al., 2016*; *Sanmartín et al., 2014*; *Vergara et al., 2007*) but only a few studies have appeared on the analysis of data obtained from HIV mono-infected individuals (*Akhtar et al., 2008*; *DallaPiazza et al., 2010*; *Han et al., 2013*; *Lui et al., 2016*; *Rivero-Juárez et al., 2013*; *Shur et al., 2016*).

With the availability of noninvasive fibrosis determinations, such as liver stiffness (LS) measurements with transient elastography, aspartate aminotransferase (AST)-to-platelet ratio index (APRI) and the FIB-4 score, cross-sectional and prospective studies to evaluate prevalence and incidence of LF in HIV-infected individuals have become easier. These tests were demonstrated to be acceptable in predicting the absence of fibrosis or mild fibrosis (LF < 2 METAVIR score) and the presence of advanced fibrosis (LF > 3 METAVIR score) (*González Guilabert, Hinojosa Mena-Bernal & del pozo González, 2010*). Cross-sectional studies in HIV mono-infected patients reported high rates (11–47%) of significant LF suggesting that HIV itself may contribute independently to liver damage (*Rockstroh et al., 2014*). Ongoing LF is not always accompanied by elevated liver enzymes. Thus, the diagnosis of LF and the prevention of progression to liver cirrhosis are important challenges. As a result, adequate monitoring strategies of liver disease are clearly needed to optimize care of HIV-infected individuals.

To date, only a few studies using LS measurements have examined the prevalence and potential risk factors for hepatic fibrosis among HIV mono-infected patients. Using different cutoff values resulted in a wide range in prevalence estimates (*Han et al., 2013*; *Merchante et al., 2010*). Pre-defined cutoffs adopted from the HIV/HCV-co-infected population may lead to an underestimation of the number of HIV mono-infected patients with clinically significant fibrosis as these cutoffs were determined for a population in which ongoing fibrosis is triggered by HCV co-infection (*Han et al., 2013*). To overcome this limitation, our aim was to use a continuous scale of LS values without any cutoff to identify significant predictors of LS in a cross-sectional study.

## MATERIALS & METHODS

### Study population

The investigation was performed in accordance with the Helsinki Declaration and was approved by the Institutional Review Board of St. István and St. László Hospital, Budapest,

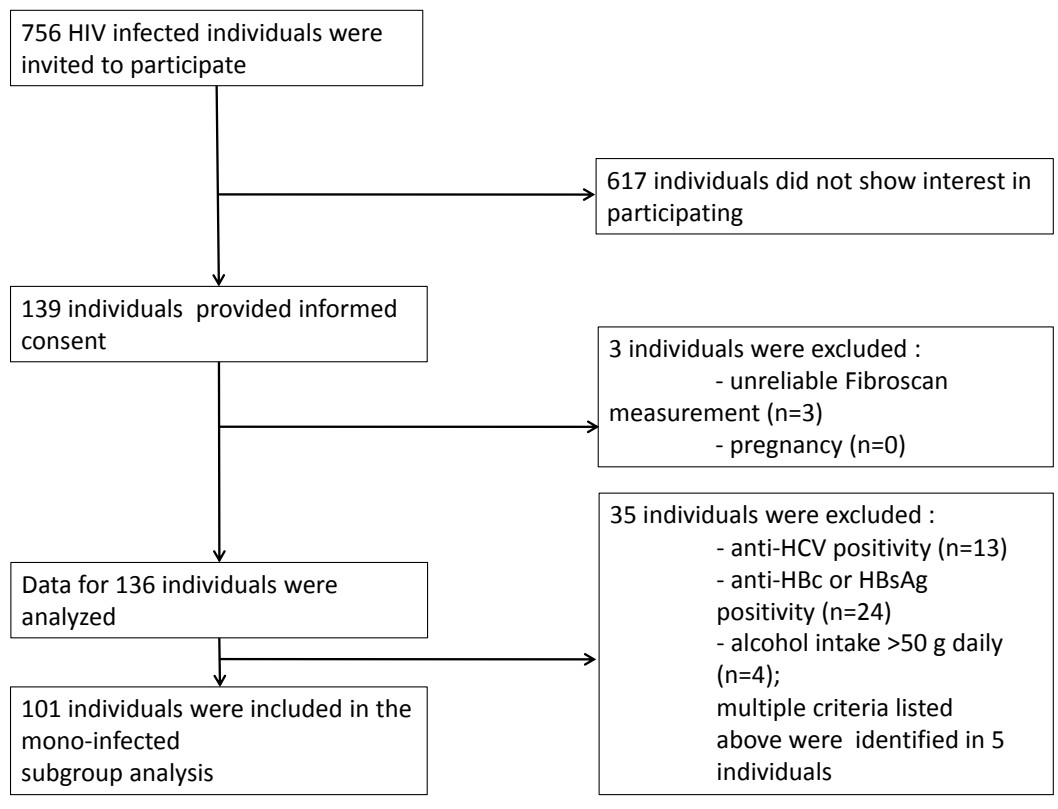

**Figure 1** Recruitment flow of the study participants.

Hungary (approval number: 34/EB/2013). Written informed consent was taken from all study participants. The present cross-sectional study is an analysis of data collected for a previous study, with methodology already described (*Sulyok et al., 2015*). Individuals older than 18 years of age were enrolled after providing their written informed consent. Pregnant women and patients with unreliable transient elastography measurement were excluded. Patients with known HCV or HBV infection or anti-HBc positivity, known other risk factors of liver diseases, or significant daily alcohol intake (>50 g/day) were excluded from the analysis.

From March 1, 2014 to October 30, 2014 all HIV-infected patients who attended the outpatient clinic at the HIV Center, St. István and St. László Hospital (Budapest, Hungary) were invited to participate in the study ($n = 756$). Liver stiffness measurements were performed on 139 patients. Out of this cohort 101 individuals were eligible for the final analysis (Fig. 1). The mode of transmission of HIV was reported to be sexual intercourse in all patients. The baseline study population characteristics are summarized in Table 1.

## Transient elastography

Transient elastography examination was performed by experienced investigators at the Hepatology Center of Buda, Budapest, Hungary, using a FibroScan 502 equipment (Fibroscan , EchoSens[TM], Paris, France). Measurements were performed using M probe on the right lobe of the liver, through intercostal spaces according to instructions by the

**Table 1  Study population ($n = 101$) characteristics.** Due to missing values descriptive statistics of BMI (body mass index) and serum triglyceride values are derived from 100 individuals, serum cholesterol, the length of known HIV positivity and hypertension from 99 individuals.

| Parameter | Mean (Median) ± SD (IQR) [Min–Max] |
|---|---|
| CD4 % | 27.6 (29) ± 9 (11) [1–46] |
| CD8% | 45.1 (44) ± 12.7 (17) [20–78] |
| CD4/8 ratio | 0.7 (0.6) ± 0.4 (0.5) [0–1.8] |
| Age (years) | 44.6 (42.4) ± 11.4 (13.4) [24.4–71.3] |
| BMI (kg/m$^2$) | 25 (24.8) ± 3.2 (3.3) [18.1–37.8] |
| Serum triglyceride (mmol/L) | 2.8 (2) ± 2.5 (2.4) [0–13.1] |
| Serum cholesterol (mmol/L) | 5.4 (5.4) ± 1.5 (1.8) [0–10.9] |
| Known HIV positivity (years) | 9.2 (7) ± 6.4 (9) [0.8–25] |
| Liver Stiffness (kPa) | 5.7 (5.1) ± 3.3 (1.7) [3–34.3] |
| CAP (dB/m) | 250.6 (239) ± 56.4 (74) [165–385] |
| | *N* (%) |
| ART ever taken | 92 (91.1) |
| Darunavir | 20 (19.8%) |
| Atanazavir | 7 (6.9%) |
| Raltegravir | 8 (7.9%) |
| Etravirine | 9 (8.9%) |
| Nevirapine | 22 (21.8%) |
| Efavirenz | 27 (26.7%) |
| Tenofovir | 38 (37.6%) |
| Abacavir | 13 (12.9%) |
| Zidovudine | 39 (38.6%) |
| Lamivudine | 89 (88.1%) |
| Lopinavir | 26 (25.7%) |
| Gender (female) | 3 (3%) |
| Diabetes | 11 (10.9%) |
| Hypertension | 21 (21.2%) |
| Lipodystrophy | 12 (11.9%) |

**Notes.**
CAP, controlled attenuation parameter; ART, antiretroviral therapy.

manufacturer. Examinations with 10 successful shots and an interquartile range (IQR) for LSs less than 30% of the median value were considered as reliable. Details of the technical background and the examination procedure have been previously described elsewhere (*Sandrin et al., 2003*). We used a continuous scale of LS values in our statistical analyses to avoid information loss emerging from categorization of the variable. However, to describe the patient population we adopted the cutoff for significant LF of 7.2 kPa and 5.3 kPa, and 14.6 kPa to define the presence of cirrhosis (*Han et al., 2013*; *Vergara et al., 2007*).

## Interview and clinical assessment

Clinical parameters were collected on the day of transient elastography examination. Recorded data were as follows: age, sex, body mass index (BMI), facial lipodystrophy assessment (defined by the presence deeper cheek atrophy), smoking, alcohol intake, drug

use, type of antiretroviral medication (ARV), co-medications, comorbidities, and date of HIV diagnosis. Biochemical and immunological parameters, blood count, CD4 and CD8 count were collected at the visit when the informed consent was obtained (<4 weeks before the LS measurement).

## Statistical analysis

The primary outcome variable was liver stiffness. The univariate association with categorical variables was assessed by a two independent sample Mann–Whitney $U$ test (i.e., Wilcoxon rank-sum test). The univariate correlation with continuous variables was assessed using the Pearson and Kendall-$\tau$ rank-correlation coefficient. Visualization was performed with scattergrams indicating best fitting linear curve and LOWESS-smoother. Holm correction was performed to counteract problems related to multiple comparisons.

Multivariate analysis was performed using Bayesian Model Averaging (BMA). Results are shown as posterior effect—or inclusion—probability (PEP), and expected value and standard deviation of the posterior distribution for each covariate (*Hoeting et al., 1999*; *Raftery, 1995*). Best models are illustrated visually by depicting the variables included in them.

Calculations were performed using R (*R Core Team, 2016*) with library BMA (*Raftery et al., 2015*). Data and script are available as Supplemental Information S1 and S2.

## RESULTS

LS ranged from 3.0 kPa to 34.3 kPa with a median value of 5.1 kPa (IQR 1.7). According to the HIV/HCV co-infection LS cutoffs, significant LF defined as LS > 7.2 kPa was detectable in 10/101 (9.9%) individuals. Presence of cirrhosis (LS > 14.6 kPa) was observed in two (1.98%) participants. Applying the cutoff (5.3 kPa) from a healthy population, significant fibrosis was detected in 45/101 (44.55%) patients.

Significant Pearson and Kendall correlation was found between LS and controlled attenuation parameter (CAP) value ($p = 0.022985$; $p = 0.0000162$), age ($p = 0.003794$; $p = 0.006593$) and BMI ($p = 0.010303$; $p = 0.000146$).With regard to categorical variables, significant association could be identified with hypertension ($p = 0.04548$) but not with ARVs. After correction due to multiple testing, only association with LS and BMI ($p = 0.0048114$) and LS and CAP ($p = 0.0005496$) remained significant. Associations of LS and different continuous and categorical variables are presented in Tables 2–3 and Figs. 2A–2I.

Next, we performed a multivariate analysis to investigate the effect of these parameters on LS. Results of BMA are given in Table 4. We identified a very high support for age (PEP: 84.5%), moderate for BMI (PEP: 49.3%), CD4/8 ratio (PEP: 44.2%) and lipodystrophy (PEP: 44.0%). On the other hand, for all remaining variables, the model rather provided evidence against their effect. Figure 3 shows the best models graphically. These results overall suggest that age and BMI have a positive association with LS, while CD4/8 ratio and lipodystrophy are negatively associated.

It is worth noting that even the best model has only 2.4% posterior probability (even the cumulative posterior probability for the 10 best models is only 15.6%).

**Table 2  Univariate analysis: associations between liver stiffness and continuous variables.** The *p*-value pertains to the null hypothesis of no correlation; *p*-values are unadjusted.

| Variable | Pearson | | Kendall | |
|---|---|---|---|---|
| | *r* | *p* | *τ* | *p* |
| CD4% | −0.087 | 0.386973 | 0.008555 | 0.901708 |
| CD8% | 0.075447 | 0.453335 | −0.01846 | 0.789103 |
| CD4/8 ratio | −0.10605 | 0.291208 | −0.00341 | 0.960177 |
| Age (years) | 0.285574 | 0.003794 | 0.185478 | 0.006593 |
| BMI (kg/m$^2$) | 0.255489 | 0.010303 | 0.26108 | 0.000146 |
| Triglyceride (mmol/L) | 0.026998 | 0.78975 | 0.079497 | 0.250808 |
| Cholesterol (mmol/L) | 0.028166 | 0.781974 | 0.059661 | 0.3915 |
| Known HIV positivity (years) | 0.147292 | 0.145703 | 0.126008 | 0.073529 |
| CAP (dB/m) | 0.226115 | 0.022985 | 0.295207 | 0.0000162 |

**Notes.**
BMI, body mass index; CAP, controlled attenuation parameter.

**Table 3  Univariate analysis: associations between the liver stiffness and categorical variables.** Liver stiffness (LS) values are presented in mean (median) ± SD (IQR) [minimum–maximum] format. *p*-value pertains to the null hypothesis of stochastic equivalence of the two populations (presence/absence).

| Categorical variable | LS in the presence of variable | LS in the absence of variable | *p* |
|---|---|---|---|
| ART ever taken | $n = 92$, 5.7 (5.2) ± 3.4 (1.8) [3.1–34.3] | $n = 9$, 4.9 (4.3) ± 1.9 (1.9) [3–9.3] | 0.13281 |
| Darunavir | $n = 20$, 5.6 (5.3) ± 1.7 (2) [3.5–10.2] | $n = 81$, 5.7 (5) ± 3.6 (1.7) [3–34.3] | 0.41051 |
| Atanazavir | $n = 7$, 5.3 (5.2) ± 1.3 (1.6) [3.6–7.3] | $n = 94$, 5.7 (5) ± 3.4 (1.7) [3–34.3] | 0.84091 |
| Raltegravir | $n = 8$, 6.2 (5) ± 3.7 (0.7) [3.9–15.3] | $n = 93$, 5.6 (5.2) ± 3.3 (1.9) [3–34.3] | 0.91481 |
| Etravirine | $n = 9$, 4.9 (4.8) ± 1 (1.9) [3.6–6.3] | $n = 92$, 5.7 (5.1) ± 3.5 (1.8) [3–34.3] | 0.42414 |
| Nevirapine | $n = 22$, 5.3 (5.3) ± 1.1 (1.8) [3.6–7.4] | $n = 79$, 5.8 (5) ± 3.7 (1.7) [3–34.3] | 0.85302 |
| Efavirenz | $n = 27$, 5.4 (5.3) ± 1.4 (2.3) [3.1–8.8] | $n = 74$, 5.8 (5) ± 3.8 (1.7) [3–34.3] | 0.59088 |
| Tenofovir | $n = 38$, 6.3 (5.3) ± 5.1 (2.1) [3.1–34.3] | $n = 63$, 5.3 (5) ± 1.3 (1.7) [3–10.2] | 0.54861 |
| Abacavir | $n = 13$, 5.6 (5.8) ± 1.8 (2) [3.6–10.2] | $n = 88$, 5.7 (5) ± 3.5 (1.8) [3–34.3] | 0.81937 |
| Zidovudine | $n = 39$, 5.5 (4.9) ± 2 (1.7) [3.7–15.3] | $n = 62$, 5.8 (5.2) ± 4 (1.9) [3–34.3] | 0.94157 |
| Lamivudine | $n = 89$, 5.7 (5.1) ± 3.5 (1.8) [3.1–34.3] | $n = 12$, 5.1 (4.4) ± 1.8 (1.9) [3–9.3] | 0.22107 |
| Lopinavir | $n = 26$, 6.7 (5) ± 6.1 (1.8) [3.6–34.3] | $n = 75$, 5.3 (5.1) ± 1.4 (1.7) [3–10.2] | 0.65209 |
| Gender (female) | $n = 3$, 4.7 (4.9) ± 0.6 (0.5) [4–5.1] | $n = 98$, 5.7 (5.2) ± 3.4 (1.9) [3–34.3] | 0.44681 |
| Diabetes | $n = 11$, 7 (6.3) ± 3.3 (2.6) [3.9–15.3] | $n = 90$, 5.5 (5) ± 3.3 (1.6) [3–34.3] | 0.06365 |
| Hypertension | $n = 21$, 6.1 (5.4) ± 2.4 (1.5) [4–15.3] | $n = 78$, 5.5 (4.9) ± 3.6 (1.8) [3–34.3] | 0.04548 |
| Lipodystrophy | $n = 12$, 5.3 (5) ± 1 (1.6) [4–6.9] | $n = 89$, 5.7 (5.1) ± 3.5 (1.7) [3–34.3] | 0.82133 |

**Notes.**
ART, antiretroviral therapy.

The best model includes age ($\beta = 0.10$ [0.039–0.16], $p = 0.00174$) and CD4/8 ratio ($\beta = -2.2$ [−4.1—0.28], $p = 0.02501$), but these results should be interpreted with caution in the light of the substantial model uncertainty.

## DISCUSSION

To our knowledge, only a few studies assessing liver stiffness in HIV-infected patients without HBV or HCV infection have been published so far. In these publications a wide

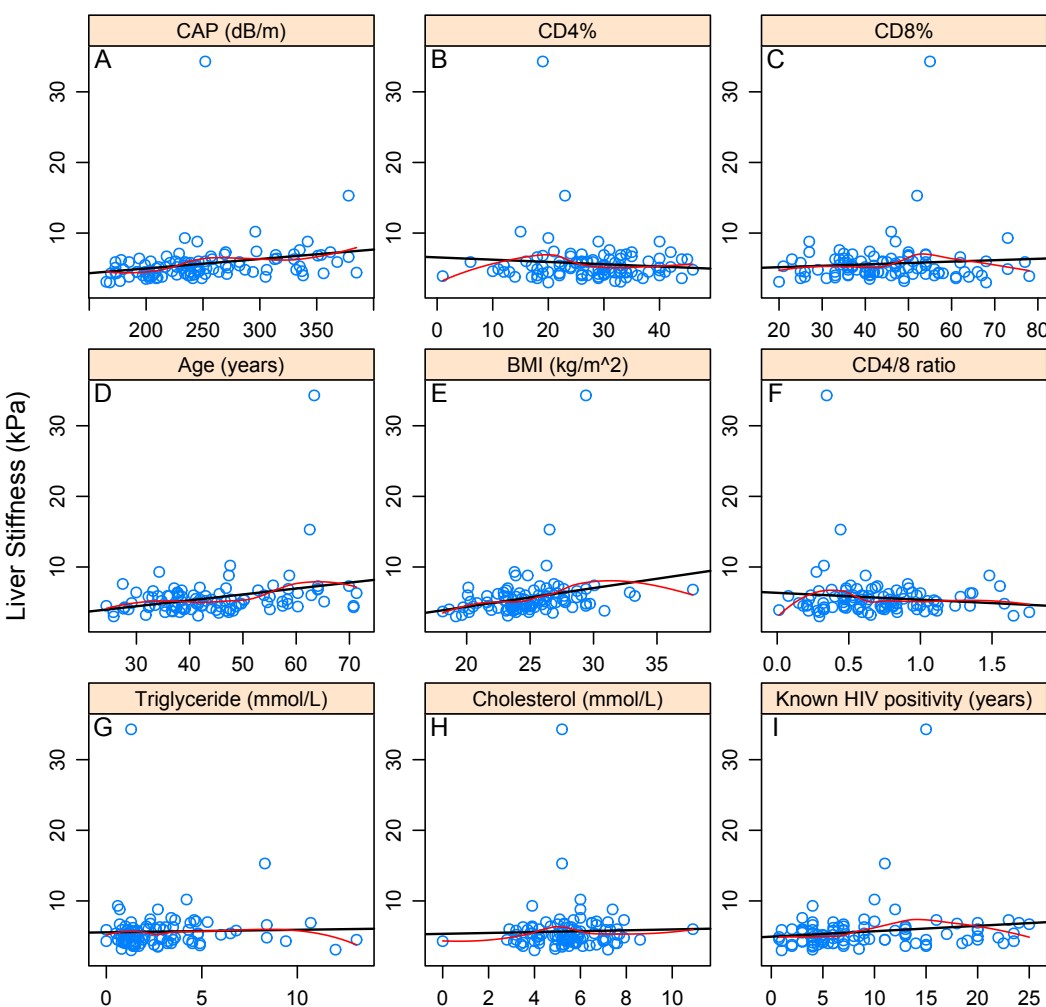

**Figure 2** **Correlations between continuous variables and liver stiffness.** The black line shows the best-fitting linear curve, the red line shows the LOWESS-smoother. (A–I) refer to the correlation between liver stiffness value and the corresponding variable.

range of prevalence for abnormal LS values were identified (*Han et al., 2013*; *Lui et al., 2016*; *Merchante et al., 2010*; *Rockstroh et al., 2014*). Using these applied cutoff values we had a similarly wide prevalence range (9.9–44.55%). These diverse results clearly underline the importance of identifying better cutoff values in HIV mono-infected patients. The most reliable method for this would be to perform liver biopsy in a large unselected HIV mono-infected population and to compare its results with those of transient elastography. Nevertheless, to our knowledge, no such study has been carried out. The discrepancies in cutoff values might lead to unreliable estimation of the rate and grade of LF. Therefore, we used a continuous scale of LS for our correlation and regression analyses to avoid uncertainty arising from using a pre-defined abnormal values as a cutoff point.

BMA revealed age as the most important predictor of LS. Age is a well-known risk factor for LF in non-alcoholic fatty liver disease (NAFLD) and HCV-infected patients (*Chan, Patel & Choi, 2016*). However, data about age-related fibrosis in the HIV mono-infected

**Table 4  Results of Bayesian Model Averaging (BMA).**

| Variables | PEP (%) | EV | SD |
|---|---|---|---|
| Intercept | 100.0 | −0.5095082 | 3.712.817 |
| CD4% | 17.1 | −0.0130202 | 0.034393 |
| CD8% | 12.9 | 0.0061567 | 0.020026 |
| Age (years) | 84.5 | 0.0827192 | 0.048982 |
| BMI (kg/m$^2$) | 49.3 | 0.1213562 | 0.147793 |
| CD4/8 ratio | 44.2 | −0.9654844 | 1.276.960 |
| Triglyceride (mmol/L) | 1.6 | −0.0005448 | 0.018470 |
| Cholesterol (mmol/L) | 1.6 | −0.0011508 | 0.030622 |
| Sex | 2.2 | −0.0277561 | 0.342417 |
| Diabetes | 3.3 | 0.0331575 | 0.282103 |
| Hypertension | 1.5 | 0.0001101 | 0.106425 |
| Lipodystrophy | 44.0 | −11.266.415 | 1.508.412 |
| Known HIV positivity (years) | 2.4 | 0.0004474 | 0.015563 |
| Darunavir | 2.0 | −0.0092738 | 0.147354 |
| Atanazavir | 1.9 | −0.0139616 | 0.231754 |
| Raltegravir | 1.5 | 0.0012294 | 0.162407 |
| Etravirine | 8.5 | −0.1378151 | 0.583234 |
| Nevirapine | 2.9 | −0.0197049 | 0.183840 |
| Efavirenz | 2.2 | −0.0102161 | 0.134693 |
| Tenofovir | 14.0 | 0.1519821 | 0.461496 |
| Abacavir | 1.6 | 0.0022876 | 0.135247 |
| Zidovudine | 10.5 | −0.1116331 | 0.406437 |
| Lamivudine | 1.5 | 0.0027499 | 0.132358 |
| Lopinavir | 26.7 | 0.3950730 | 0.778517 |
| CAP (dB/m) | 8.9 | 0.0009069 | 0.003627 |

**Notes.**
PEP, Posterior effect probability; EV, expected value of the posterior distribution of the parameter; SD, standard deviation; CAP, Controlled attenuation parameter; ART, Antiretroviral therapy.

population are scarce (*Rockstroh et al., 2014*). To date, only a few descriptive studies identified significant association with age and LF in this patient population (*Blanco et al., 2011*; *Han et al., 2013*; *Merchante et al., 2010*). Ageing has multiple effects on the liver, making it more vulnerable to fibrogenetic factors. The exact mechanism, however, remains unknown. The decreased regenerative capacity, microbial translocation and HIV-induced immunologic dysfunction as well as chronic inflammation may play non-mutually exclusive roles (*Chan, Patel & Choi, 2016*). This result was also in line with our other finding, the identified remarkable negative association between LS and CD4/8 ratio. The low CD4/8 ratio is an accepted marker of HIV-induced immune dysregulation (*Serrano-Villar et al., 2014*). Therefore, this observation could reflect on the role of HIV-induced immune dysregulation in the development LF. In this population, persisting abnormally low CD4/8 ratio is associated with impaired gut mucosal immunity (*Serrano-Villar et al., 2014*). Destruction of the mucosal barrier leading to microbial translocation could be a driving force of LF. In a recent study, a marker of microbial translocation, elevated sCD14

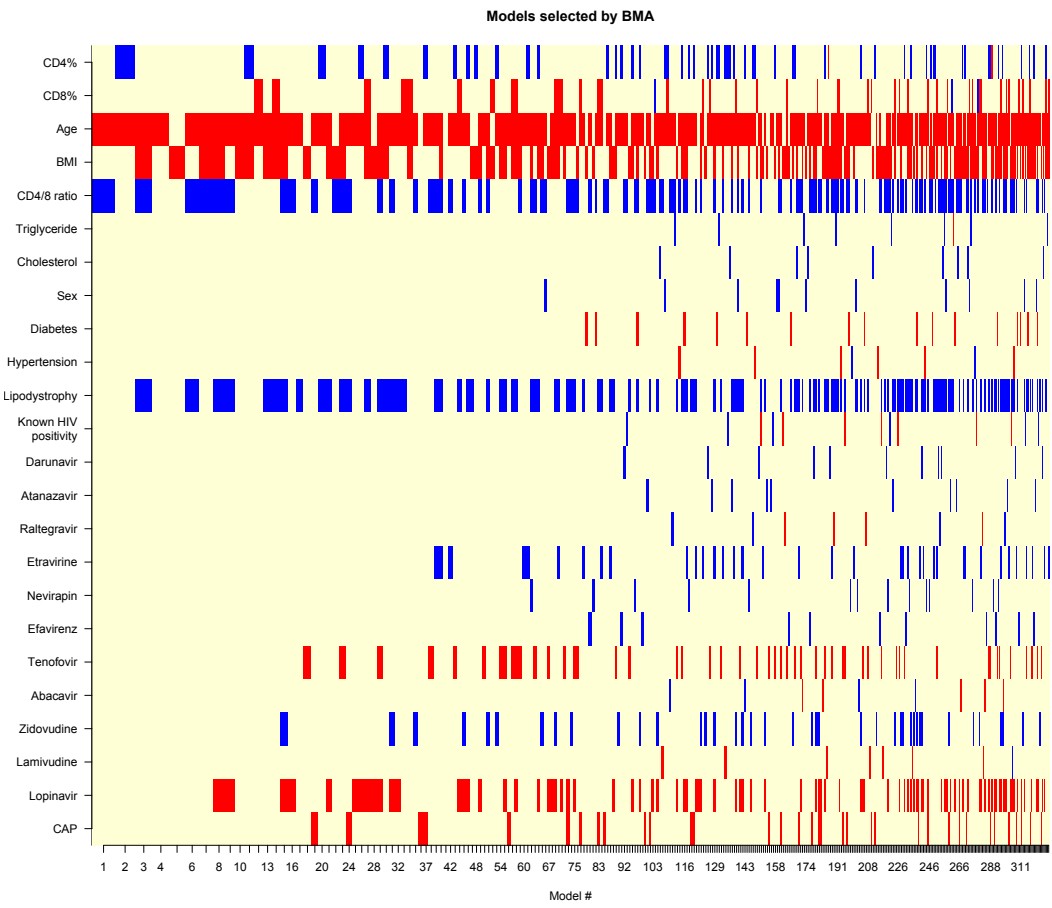

**Figure 3** **Models selected by BMA (Bayesian Model Averaging).** Red color displays negative, blue displays positive variable estimate (uncolored variables were not included in the model). On the *x*-axis, models are listed in the order of decreasing posterior model probability.

levels were associated with increased LS in HIV mono-infected individuals (*Redd et al., 2013*). In HCV-infected patients, the CD4/8 ratio as a contributing factor to LF has also been considered (*Feuth et al., 2014*). Furthermore, CD4 cells can stimulate anti-fibrotic natural killer cell activity, therefore, loss and impaired activity of CD4 cells may contribute to the progression of LF (*Rockstroh et al., 2014*). Data suggesting HIV-induced effects on the pathogenesis of fibrosis generation has been described mainly in patients with HIV/HCV co-infection (*Rockstroh et al., 2014*) but the mechanism has still not been exactly determined. In context of the ageing HIV population, a better understanding of how ageing interacts with HIV-induced immunologic and metabolic changes will have paramount importance in reducing the burden of liver diseases (*Chan, Patel & Choi, 2016*).

CAP value, quantifying hepatic steatosis showed significant correlation with LS in the univariate analysis. Remarkably, NAFLD is the most frequent cause of liver damage in this population (*Rivero-Juárez et al., 2013*). However, other studies found no association with LS and CAP (*Macías et al., 2014*; *Macías et al., 2016*). Nonetheless, multivariate analyses rather provided evidence against the effect of CAP on LS.

The correlation between BMI and LS portrays a similar profile. BMI, the most important predictor of CAP value in the HIV-infected population (*Macías et al., 2014*; *Macías et al., 2016*; *Sulyok et al., 2015*) showed significant association with LS in the univariate analysis. This association remained considerable according to the result of the BMA. This suggests, that obesity may have an independent unfavorable effect on LF even in the absence of -with CAP detectable- hepatic steatosis.

No significant association was observed between LS and ARVs. These results underline the importance of antiretroviral treatment, however, other studies have raised questions about the role of older ARVs in LF development. A cumulative exposure to boosted protease inhibitors (PI) was identified as a significant independent negative predictor of LF (*Han et al., 2013*). A possible explanation of this result could be, that a longer cumulative boosted PI exposure may reflect on a better long-term control of viral load and a lower grade of immune dysregulation. Since body-fat composition abnormalities are associated with PI exposure (*Grinspoon & Carr, 2005*), the identified negative association with the presence of facial lipodystrophy in our study may further support this theory. However, prospective, controlled trials are clearly warranted to clarify the role of PI therapy in the development of LF. Associations with didanosine and stavudine with hepatic fibrosis were previously described (*Akhtar et al., 2008*; *Blanco et al., 2011*; *Merchante et al., 2010*). In our investigated population, the number of dideoxynucleoside exposed patients was negligible ($n = 2$); therefore, we did not include these ARVs in our analysis.

The observed outlier value in one participant (LS = 34.3 kPa) refers to an advanced liver disease of unknown origin. Similarly, other observational studies in the HIV mono-infected population also identified individuals with high grade fibrosis and even with cryptogenic cirrhosis (*Lui et al., 2016*; *Merchante et al., 2010*). Recently, cirrhosis was identified in 5.2% percent of the HIV mono-infected patients (defined as LS > 10.3 kPa) compared to the 0.6% of the uninfected control group (*Lui et al., 2016*). These data underscore the importance of identifying other underlying liver diseases and improving the understanding of pathomechanism.

It is worth contrasting these result with those obtained using traditional linear regression (without variable selection). At 5%, age ($p = 0.0415$), BMI ($p = 0.0204$), presence of lipodystrophy ($p = 0.0131$), history of taking zidovudine ($p = 0.0442$) and lopinavir ($p = 0.0173$) were significant. However while this model has an apparent $R^2$ of 36%, its realistic—overfitting-optimism corrected—$R^2$ is practically zero (obtained through bootstrap validation). Thus, regularization was applied—with the penalty parameter selected by Hurvich and Tsai's corrected AIC—which resulted in a realistic model, however, it had no significant variable at all (*Harrell, 2016*). This experiment clearly illustrates the problems of modelling with so limited sample size, and the possible advantages of BMA. In particular for small datasets the effect of model uncertainty can be substantial—this is disrespected in the framework of traditional regression modelling. Variable selection is often employed; however, when it is non-blinded to the outcome, it leads to models that are biased in virtually all of their parameters. For small sample sizes, the sound alternatives - such as regularization - might lead to results that are clinically not meaningful. BMA is a relevant alternative, which avoids these issues by explicitly considering many models.

Our study has considerable limitations. The observational nature and low patient number being probably the most important ones. The number of excluded patients with significant alcohol intake has also to be dealt with caution. Since alcohol consumption was assessed by self-reporting, there is a possibility that not all affected individuals were identified. Moreover, the distance between the HIV center, where screening occurred and the hepatology center where transient elastography measurement took place was the main reason potential participants refused participation in the study. This could lead to selection bias, since low-compliance patients could be underrepresented in the study population.

## CONCLUSIONS

In conclusion, using previously described cutoff values we identified a high prevalence of hepatic fibrosis in HIV mono-infected patients. Our findings shed light on the relevance of HIV-induced immune dysregulation and overweight in the ageing HIV-infected population. The negative association between LS and the presence of lipodystrophy may reflect on the protective effect of prolonged exposure to antiretroviral therapy.

Fueled by the ongoing silent epidemic of obesity, the burden of liver diseases in individuals living with HIV shifts away from viral hepatitis coinfections to the NAFLD spectrum. A better understanding of factors leading to fibrosis will be the cornerstone of reduction in liver-related disease burden in the HIV-infected population. Nonetheless, further controlled studies are warranted to clarify causal relations.

## ACKNOWLEDGEMENTS

We are indebted to Erzsebet Varga, Kornelia Barbai, and Agnes Kissne Halasz for data collection and organization. We are also immensely grateful to Fiona O'Rourke for language correction.

### Funding

Tamás Ferenci was supported by the New National Excellence Program of the Ministry of Human Capacities, Hungary (UNKP-16-4/III ). The funders had no role in study design, data collection and analysis, decision to publish, or preparation of the manuscript.

### Grant Disclosures

The following grant information was disclosed by the authors:
New National Excellence Program of the Ministry of Human Capacities, Hungary: UNKP-16-4/III.

### Competing Interests

MS has been an investigator in clinical trials supported by Novartis, Bristol-Myers Squibb, Janssen-Cilag, Actelion, and Abbvie Pharmaceuticals. GH has served as a consultant and/or an investigator for and received consulting/speaker fees from Abbott, Abbvie, Boehringer-Ingelheim, Bristol-Myers Squibb, Fresenius-Kabi, Janssen-Cilag, Merck Sharp & Dohme,

Novartis, Pfizer, and Roche. MM has been an investigator in clinical trials supported by Novartis, Bristol-Myers Squibb, Janssen-Cilag, Abbvie, Roche, Boehringer-Ingelheim, and Merck Sharp & Dohme. He has received lecture and consultant fees from Janssen-Cilag, Abbvie, Roche, Boehringer-Ingelheim, Merck Sharp & Dohme, and Gilead. JS has been an investigator in clinical trials supported by Novartis, Bristol-Myers Squibb, Janssen-Cilag, Pfizer, Merck Sharp & Dohme, and Abbvie Pharmaceuticals. He has received lecture and consultant fees from Janssen-Cilag, Abbvie, Roche, Boehringer-Ingelheim, and Merck Sharp & Dohme. For the remaining authors there are no conflicts of interest.

## Author Contributions

- Mihály Sulyok and Zita Sulyok conceived and designed the experiments, performed the experiments, analyzed the data, wrote the paper, prepared figures and/or tables, reviewed drafts of the paper.
- Tamás Ferenci analyzed the data, wrote the paper, prepared figures and/or tables, reviewed drafts of the paper.
- Mihály Makara and Zsófia Rupnik performed the experiments, wrote the paper.
- Gábor Horváth performed the experiments, contributed reagents/materials/analysis tools, wrote the paper.
- János Szlávik, Luca Kormos and Zsuzsanna Gerlei contributed reagents/materials/analysis tools, wrote the paper.
- István Vályi-Nagy conceived and designed the experiments, wrote the paper, reviewed drafts of the paper.

## Human Ethics

The following information was supplied relating to ethical approvals (i.e., approving body and any reference numbers):

The investigation was performed in accordance with the Helsinki Declaration and was approved by the Institutional Review Board of St. István and St. László Hospital, Budapest, Hungary (approval number is 34/EB/2013).

## Data Availability

The raw data has been supplied as a Supplemental File.

## Supplemental Information

Supplemental information for this article can be found online at http://dx.doi.org/10.7717/peerj.2867#supplemental-information.

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
