# Peer review of "Hepatic fibrosis and factors associated with liver stiffness in HIV mono-infected individuals"

_PeerJ, doi:10.7717/peerj.2867_

## Round 0.1 · original submission · Major Revisions

The authors describe an epidemiological survey of HIV mono-infected patients on liver fibrosis in Hungary. They tried to search for factors having effect on liver stiffness. Transient elastography is one of the most noninvasive methods for staging liver fibrosis.

In this type of experiments, it is important to be careful to deal with the factors, which inherently associated with progression of hepatic fibrosis, because ex. CAP, BMI, or diabetes are associated with progression of liver stiffness. In addition, the number of patient receiving each antiviral compound (8 patient received ART, combination of 12 drugs) may not enough to lead statistically significant conclusions.

Also it should be noticed that abacavir is sometimes associated with side effect on liver. Conclusions could be obtained from the ordinary results. It is desirable for the authors to upload the raw data for revise.

Liver stiffness value did not reveal correlation with CD4/CD8 ratio or disease duration (progression of virus disease) meaning that HIV-mono infection does not associated liver stiffness. Taken together the authors could change the conclusion that the factors examined in this manuscript were not associated with progression of liver fibrosis. Alternative conclusions could be changed to use clinical data to confirm no relationship of the factors with liver stiffness among HIV-mono infected individuals.

The format of the manuscript is unfortunately not sophisticated. For Materials & Methods should rewrite with short headings, and describe concisely. If you don't explain “transient elastography examination”, proper references should be cited for understanding, and add description about cutoff value of the method.

The authors need to describe Results in details, and Discussion is redundant, should be concisely written.

·

Basic reporting

Minor specific comments:

1. Introduction, line 51-53. You should show the papers have been published.

2. Discussion, line 149, 172, 209 and 211. You say “LF”. Does it mean liver fibrosis?

Experimental design

No Comments

Validity of the findings

The manuscript by Zita Sulyok et al evaluates the degree of hepatic fibrosis in HIV-mono-infected patients. Interestingly, the presence of diabetes, age, CAP value and a history of taking abacavir were positively correlated with higher values of liver stiffness. Taking etravirine was negatively associated with liver stiffness, but taking abacavir was positively associated as previously reported. It would be important knowledge for medications and treatment strategies against liver damage in HIV-infected patients.
The data of correlations between continuous variables and the liver stiffness is relatively clean and convince. The writing of the manuscript is overall clear and easy to follow.

Major comments:

1. You show significant correlation between diabetes and liver stiffness (Fig. 3) and say the importance of diabetes in the development of LF which can be triggered by ongoing HIV replication (discussion, line 171). But I could not figure out that diabetes positively correlate with liver stiffness in HIV-infected patients since you did not perform in non-HIV-infected patients. If you say some additional information about the correlation between diabetes and liver stiffness in the non-HIV-infected patients, it will help for our understanding.

Reviewer 2 ·

Basic reporting

1. In Abstract, line31-33, there are no descriptions of methods in this study. (ex. Transient elastography (FibroScan [FS]) was performed to measure liver stiffness.)
2. In Abstract, line34, “LSMs” could changed into “liver stiffness measurements (LSMs)”.
3. In Abstract, line 35, “CAP” could be changed into “controlled attenuation parameter (CAP)”.
4. In results, line 125, p values are not identical shown in Table 2 for example, “body mass index ( p = 0.01)”
5. In results, line 131, “value (p=0.0061), age (p=0.04356)” are different values from Table4.
6. Figure 2 and Table 2, if possible, please add a variable “known HIV positivity” in Table 2 additionally.
7. Figure 2, the dot of stiffness value “34” could not be found in ART exp. in Figure2. Are the plotted scattergrams involving all of the values in this study?
8. Figure 3 and Table 4, the list of variables in Figure 3 and Table 4 was not ordered in the same turn.
9. Table1, “ARTb ever taken 125” here what is the meaning for “125” ?

Experimental design

In this study, only HIV-mono-infected individuals were recruited for the basic analysis. No data or cited references described about how the indicated variables, CAP, Age, diabetes, are contributing to liver stiffness in healthy individuals.

Validity of the findings

The non-invasive imaging technique in this study used to measure liver stiffness is considered to be very helpful for the evaluation of liver disease development, and it affords the chance that patient are spared from histologic assessment. LS cutoff values assessed by means of TE for predicting liver disease stage varied depending on the different populations. Therefore, author used the continuous scale of LS for correlation and regression analysis. In conclusion, in case of HIV-mono-infected individuals, Age, diabetes and CAP were the important factors associated with the liver stiffness.

Additional comments

In this study, only 8 individuals of 101 participants took 11 antiviral drugs, including Darunavir, Atanazavir, Raltegravir, Etravirine, Nevirapin, Efavirenz, Tenofovir, Abacavir, Zidovudine, Lamivudine, and/or Lopinavir. The ART-exp. subpopulation looks limited, if possible, please list N (%) for every compound in Table1.

---

## Round 0.2 · Minor Revisions

The authors performed a major revision of their manuscript, including correction of the dataset and the multivariate model. Although a significant statistic value was not obtained by univriable analysis with dataset of the 101 HIV mono-infected patients, through Bayesian model averaging (BMA) method, there was moderate negative association between hepatic stiffness and CD4/8 ratio (Posterior Effect Probability 44.2%). It is important to define effect of HIV infection and progression on hepatic fibrosis. The manuscript should be accepted for publication with the following minor revisions.

1. As the reviewer points out, add comment on the outlier value (LS value, >30 in Fig. 1).
2. Move the units into Figure 1 from the legend including triglyceride, cholesterol, and known HIV positivity (should be "year").
3. Add some comments on an advantage of Bayesian model averaging (BMA) compared with other models or analytical approaches in “Results” or “Discussion”.

·

Basic reporting

The authors have done a good job at answering most of the reviewers' several points. In my opinion, the authors' manuscript is publishable.

Experimental design

No comments

Validity of the findings

No comments

Reviewer 2 ·

Basic reporting

Figure2, the unit of y-axis and x-axis is not clearly marked.
Table 3, Results are presented in mean (median)… Does the word “Results” means LS values?
Table 3, …two populations (presence/absence). CAP:controlled attenuation parameter; ART: antiretroviral therapy. Here, “CAP:controlled attenuation parameter;” could be removed.

Experimental design

No Comments.

Validity of the findings

No Comments.

Additional comments

No Comments.

---

## Round 0.3 · accepted · Accept

The authors added explanation for the outlier value and for an advantage of BMA in Discussion. The manuscript has been revised well, and should be accepted for publication.